# Crowdsourcing as a Tool for Urban Emergency Management: Lessons from the Literature and Typology

**DOI:** 10.3390/s19235235

**Published:** 2019-11-28

**Authors:** Ramon Chaves, Daniel Schneider, António Correia, Claudia L. R. Motta, Marcos R. S. Borges

**Affiliations:** 1Programa de Pós Graduação em Informática, Federal University of Rio de Janeiro (PPGI/UFRJ), Rio de Janeiro 21941-590, Brazil; ramon.mchaves@gmail.com (R.C.); claudiam@nce.ufrj.br (C.L.R.M.); 2Tércio Pacitti Institute of Computer Applications and Research (NCE/UFRJ), Rio de Janeiro 21941-916, Brazil; 3INESC TEC and University of Trás-os-Montes e Alto Douro (UTAD), 5001-801 Vila Real, Portugal; 4School of Computing, University of Kent, Canterbury CT2 7NT, UK; 5School of Engineering at San Sebastián, TECNUN, University of Navarra, Donostia, 20018 San Sebastián, Spain

**Keywords:** crowdsourcing, emergency management, quality control in crowdsourcing, systematic literature review, task design, urban planning

## Abstract

Recently, citizen involvement has been increasingly used in urban disaster prevention and management, taking advantage of new ubiquitous and collaborative technologies. This scenario has created a unique opportunity to leverage the work of crowds of volunteers. As a result, crowdsourcing approaches for disaster prevention and management have been proposed and evaluated. However, the articulation of citizens, tasks, and outcomes as a continuous flow of knowledge generation reveals a complex ecosystem that requires coordination efforts to manage interdependencies in crowd work. To tackle this challenging problem, this paper extends to the context of urban emergency management the results of a previous study that investigates how crowd work is managed in crowdsourcing platforms applied to urban planning. The goal is to understand how crowdsourcing techniques and quality control dimensions used in urban planning could be used to support urban emergency management, especially in the context of mining-related dam outages. Through a systematic literature review, our study makes a comparison between crowdsourcing tools designed for urban planning and urban emergency management and proposes a five-dimension typology of quality in crowdsourcing, which can be leveraged for optimizing urban planning and emergency management processes.

## 1. Introduction

Emergency management in urban environments is marked by the complexity in the decision-making processes that require a dense set of relationships between multiple stakeholders [1,2]. Among a wide range of possible urban disasters, which can affect millions of people in the closer cities, this paper explores the dam rupture example to elucidate how crowd workers can be leveraged as human sensors and distributed intelligent systems able to work on complex problems. The term *crowdsourcing* was originally presented as a web-based business model, where solving a problem is made possible through the voluntary collaboration of a crowd of digital workers [3]. As pointed out by Correia et al. [4], different forms of digital participation and public engagement can be established in large-scale projects that harness the crowd workforce [5]. Accordingly, crowdsourcing approaches have been supported by a range of technology-mediated platforms that rely on the wisdom of crowds to generate innovative solutions in urban planning processes. In this perspective, as long as crowdsourcing platforms are designed to motivate public participation, a crowdsourcing approach could offer numerous advantages compared to traditional face-to-face meeting models [6].

In recent years, citizen involvement has been increasingly used in disaster prevention and management, taking advantage of new pervasive and ubiquitous technologies. This new scenario has created a unique opportunity to leverage the work of crowds of volunteer citizens. As a result, a set of computational tools based on crowdsourcing and *mobile participatory sensing* have been proposed and evaluated. Looking at the insights reported in a standard document released by the National Water Agency [7], 3.5 million people live in 45 cities with risk of dam disruption in Brazil. Even considering the scenario in which many companies conduct environmental impact studies and risk reports, disasters continue to occur. Many of them are due to the negligence of employees or managers, poor facility infrastructure, or lack of maintenance. As a result, the lack of supervision by competent public structures and the negligence of corporate decision-making cause irreparable losses. Entire cities are in danger of disappearing from the map, including Congonhas, which is a UNESCO World Heritage Site since 1985. Drawing from the National Water Agency’s observational study of urban water supply and pollution control [7], dams with a high degree of danger are situated in the Belo Horizonte Metropolitan Region, the second largest state population. Such dams have chemical waste from gold mining. In the case of rupture, waters will be contaminated with cyanide, a substance with high toxicity. Environmentalists remain on alert for the proximity of these dams to Rio das Velhas, a river accountable for supplying fresh water to one-third of the population of Belo Horizonte, besides being one of the branches of the São Francisco River that is another important source of fresh water. Unfortunately, this is only one of the many challenges that individuals and communities may face. This means that we urgently need new approaches to prevent and deal with new occurrences of dam disruptions.

In order to reduce the risk of disasters through systematic efforts of prevention and early detection, the concept of *disaster risk reduction* [5] has been explored in the literature. The idea is to adopt diverse available instruments and strategies at different stages of the disaster process, including disaster prediction, community awareness by means of disaster risk education, and establishment of early warning systems [8,9,10]. Despite previous efforts and initiatives to address the problem, a consistent reduction in current disaster risk levels compared to the 1990s has not been verified [11]. The main needs in urban management for those threatened cities are: (i) Mapping for emergency plans; (ii) route escape simulations; and (iii) emergency shelter and settlement design [12]. Respectively, the following techniques are addressed as solutions: Mobile crowdsourcing, volunteered geographic information (VGI), and tournament crowdsourcing (TC). Furthermore, in light of these findings, aspects related to the quality of crowdsourcing are further investigated.

We claim that crowdsourcing approaches may bring clearly identifiable benefits when applied to this domain, as disaster management tasks are difficult to automate due to their heterogeneity and inconsistency [13]. Moreover, crowdsourcing approaches for urban disaster prevention and planning have the potential to enable a crowd of citizens to asynchronously build and store knowledge about many issues and actively participate in decision-making by engaging in digital tasks. However, this organization and articulation of citizens, tasks, and outcomes, as a continuous flow of knowledge generation, reveals a complex ecosystem that requires coordination efforts to manage interdependencies in crowd work [14,15]. To address these shortcomings, this paper extends the results of a previous study [16], which investigates how crowd work is managed in crowdsourcing platforms applied to urban planning. Our findings shed light on the fact that many of them have a diverse set of specific urban contexts that are similar to emergency management when considering socio-technical spatial problems. Thus, we synthesize the results of the original article and the following research question has been added in the discussion: How can crowdsourcing techniques and quality control dimensions used in urban planning be used to support the urban emergency management in mining-related dam disruption?

The remainder of this paper is structured as follows. Section 2 introduces the motivation for this work and discusses some related literature on quality control in crowdsourcing. Section 3 presents the methodological guidelines adopted in this systematic review to characterize work management in crowdsourcing platforms when applied to urban planning. The results of this study are provided in Section 4. Then, Section 5 comprises the main results achieved in this study in order to understand how the lessons learned in crowdsourcing for urban planning can be applied to the context of urban emergency management. Finally, Section 6 concludes the paper, discusses its limitations, and points to future work directions.

## 2. Background and Motivation

### 2.1. Dimensions of Quality Control in Crowdsourcing

Successively reaching high-quality outputs from a massive number of crowd participants is still a challenge for the widespread adoption of crowdsourcing systems [17]. Here, the definition of Crosby [18] is adopted, where the notion of quality is understood as “conformity to requirements”. That is, in order for crowd work to be considered to be of high quality, it is certainly necessary that proposed urban planning tasks should be performed according to the aforementioned requirements. We characterize the quality of crowd work as the product of two dimensions: (i) The design of the tasks, and (ii) the profile of workers [1,19]. Given these two aspects, further consideration needs to be given to the following issues: Which tasks the crowd can currently perform; their choice of tasks; how task results are used; the level of complexity of tasks; the rewards and strategies for engagement; the evaluations of work performed; and the reputation of workers. In this sense, the quality of crowd work outputs is largely influenced by aspects like the abilities of crowd workers, task descriptions, incentive mechanisms and motivational strategies (e.g., gamification), level of interaction between requesters and crowd workers, software (or platform), and processes used to detect deviant crowd behavior [19]. Thus, there is a recurrent need for common rules and novel strategies to cope with quality issues in crowdsourcing for urban emergency scenarios due to their uncertain and highly dynamic nature.

In the literature, several models have been proposed for dealing with quality control and management issues in crowdsourcing. For instance, Zheng and co-workers [20] proposed a model that relies on the influence of task difficulty on the crowd worker’s abilities and general outputs. Ho et al. [21] conducted a series of experiments and proved that, depending on the type of task being performed, performance-based payments can have a positive effect on the quality of crowd work. A bibliometric analysis aimed at examining the implementation of crowdsourcing in urban planning [22] identified quality control as one of the key conceptual developments since 2015. This is line with our results presented in Section 4. In addition, such fact is also corroborated by the work of Lukyanenko and colleagues [23]. According to the authors, information quality is a core problem in citizen science endeavors. One of the main challenging issues in large-scale data collection is the unavailability of reliable methods for detecting errant behaviors (e.g., data sabotage) while maintaining high participation rates. Aside from the inherent concerns that arise when participating in citizen science projects, it is critical to ensure the equal treatment of participants with different levels of expertise as sources of potential information. For instance, embedded assessments are among the many features that can be incorporated into crowdsourcing systems for improving data quality. Closely related to this evolving strand of research is the need of developing mechanisms for identifying and assessing the “right” participants for a certain crowdsourcing task based on their prior background (e.g., skills/expertise, reputation, record of completed tasks), interests, and availability [24,25]. Simultaneous to these advances in the crowdsourcing literature, other work has emerged that attempts to use gamification approaches in the evaluation of water level class observations. In this regard, CrowdWater [26] was introduced as a system with gamified elements for measuring the accuracy of crowdsourced data.

The use of wikis as *stigmergic* approaches for geographical data collection (e.g., land cover maps) has been also addressed as a successful way of engaging crowds of citizens to improve the data quality [27]. According to Sideris and colleagues [28], some of the major challenges faced by urban planning experts are established on the reliability, validity, and standardization of data sources. However, the literature on these concerns is limited. In particular, only a few of the citizen science projects had a dedicated strategy to deal with these issues in urban emergency management and disaster relief scenarios [29,30]. Some of the core challenges that remain unsolved in the context of crowdsourced social media data for disaster management (e.g., flood mapping) include data quality and accuracy (e.g., false reports) [31], and lack of coordination among disaster relief institutions [15]. Another major concerns are the lack of trust management instruments, unequal access to citizen science tools and resources, privacy and ethical issues, hesitancy of experts when relying on the outputs provided by amateurs (citizen scientists), and absence of mechanisms for predicting the accurate allocation of resources in accordance with the contextual elements (e.g., locations, time) of each disaster relief process [32].

### 2.2. Main Challenges for Cities in Urban Emergency Management of Tailing Dam Failures

#### 2.2.1. Mapping in Support of Emergency Plans

Due to recent tragedies and the resulting changes in Brazilian legislation, it is urgent that companies responsible for mining dams update their emergency action plans to be sent to city authorities and civil defense agencies. Among other information, a dataset should be maintained, including:(i)All dwellings and sirens that cover the floodplain—self-saving zone (ZAS)—informing the number of dwellings and people affected;(ii)The meeting points in ZAS, informing the escape route, with the number of people expected at each point;(iii)A complete map of the dam break covering ZAS and self-rescue zone (ZSS), informing the total of affected people (ZAS and ZSS), as well as identifying within the floodplain the existence of sensitive buildings (schools, hospitals, health centers, daycare centers, barracks, police stations, forums, and prisons);(iv)Federal, state, and urban highways with large circulation of vehicles that will need to be banned;(v)Identify the routes that should be used as detours, block points and alternative routes.

Although many research studies have examined the use of crowdsourcing in urban planning, few studies of current factors in developing flood risk mitigation strategies could be found in the literature. One notable exception is the work of Cheung and Feldman [33] who examined the use of citizen science for mitigating flood hazards. Following this line of thought, Li and co-authors [34] called for more research into water ecological environment protection. Researchers following this path have proposed crowdsourcing tools like Crowdhydrology [35] and Social.Water [36] as complementary approaches for collecting stream water levels. Consistent with this suggestion, Gharesifard and colleagues [37] addressed the potential of community-based monitoring and citizen science for supervising and assessing data related to environmental and hydrological resources.

#### 2.2.2. Route Escape Simulations

After the Mariana dam disaster in 2015 [38], changes in Brazilian legislation have forced dam companies to support and participate in emergency drills, jointly with city halls, civil defense agencies, the dam safety team, and the population comprised in the ZAS. There is also a requirement to install sirens and other warning mechanisms. The State Coordination of Civil Defense of Minas Gerais, the state where the municipality of Mariana lies, developed a methodology for the preparation and performance of simulated exercises aimed at the population covered by ZAS in order to support cities, communities, and entrepreneurs in risk management actions. It was thus possible to evaluate the results obtained, which facilitated the correction of gaps in the process of mobilization and community participation in the exercises performed.

The aim of the simulation is to enable the entire population in the risk area to become aware of escape routes and meeting points (Figure 1). Once the sirens are turned on, people should head to the rendezvous points via previously designed escape routes. Located at strategic points, support teams timed the arrival time, and at the end of the simulation, participants answered a questionnaire to assess their opinion and measure the effectiveness of the simulated exercise.

#### 2.2.3. Emergency Shelter and Settlements Design

In situations of major urban disasters, the need for rapid rebuilding of houses, infrastructure, and public facilities is a major challenge in a stressful and resource-constrained environment. Therefore, the use of emergency shelters is an operation of transferring the unsheltered to prepared temporary structures that present area availability for such purpose. They must have an infrastructure capable of meeting the basic needs of water and sanitation, among others, ensuring the survival of those affected. Assembly of the shelters should be done within a short time (a couple of days or weeks). As stated by Costa and colleagues [39], these structures are more complex and need to be able to promote quality humanitarian assistance.

As it can be observed from Figure 2, shelters can be immediate, which are usually light and industrialized structures such as synthetic membranes and metal structures, or temporary as intermediate constructions between emergency (immediate) shelters and the reconstruction of homes, often using local materials and community participation in their construction [40].

However, the challenges regarding shelter deployment begin at the preparation stage. Urban, infrastructural, and architectural projects are necessary to ensure better accommodation of the homeless and to bring dignity throughout the different stages of emergency management. The choice of location, the techniques used for building shelters, as well as the community equipment used can be predicted in the projects, despite the uncertainties. These design solutions must be specific to each type of disaster and the case of dam disruption would be no different. In addition, the particularities of each affected city or neighborhood must be observed, which means that despite generic project solutions for the impacts of the dam disruption, specific projects should be developed for the different parts of the territory, considering topography, possible areas of impact, infrastructure, population characteristics, and so on.

## 3. Materials and Methods

A systematic literature review can be understood as a “form of secondary study that uses a well-defined methodology to identify, analyze and interpret all available evidence related to a specific question in a way that is unbiased and (to a degree) repeatable” [41].

This section describes how the literature review was conducted in order to characterize work management on crowdsourcing platforms when applied to urban planning. The review was performed according to the PRISMA Statement [42], which includes a set of items that must be considered when completing a systematic literature review. In addition, the study was based on the quality taxonomy for crowdsourcing systems [43] in order to formulate the research questions and the classification criteria presented in previous work [44].

### 3.1. Research Questions

The planning of this review allows us to identify goals and build a protocol that presents the research methodology used. This leads to the following research question (RQ):

(RQ1) How does crowd work function in crowdsourcing platforms intended to support urban planning?

More specifically, how does the management of crowdsourcing processes take place, and how does the management and control of the tasks that arise operate? To help answer this, the RQ was broken into a set of research questions that were directly associated with the quality dimensions of work on crowdsourcing platforms [1].

(RQ1.1) Task definition: What type of tasks are requested from participants?

(RQ1.2) User interface: How do participants choose their tasks on the interface?

(RQ1.3) Granularity: Do the tasks performed show traces of complexity, such as interdependence and coordination?

(RQ1.4) Policy compensation: What kind of strategies and motivations for engagement can be identified?

(RQ1.5) Work profile: To what extent were issues of work quality, performance, and worker expertise/reputation assessed?

The review introduces three auxiliary attributes that are intended to categorize the platform, including: (i) The name of the platform; (ii) the urban planning context; and (iii) the crowdsourcing technique applied. As a result of a revision of a prior version of this paper [16], we made some adjustments to our original protocol. 

### 3.2. Search Process

The search string was developed through the combination of terms that capture crowdsourcing applications oriented to urban planning or urban design. Some tests were carried out on Google Scholar to evaluate the quality and quantity of the returned results. Search terms were included in both English and Portuguese. Afterwards, we used the boolean operator OR to link the main terms. Moreover, all search terms were combined using the boolean operator AND. Finally, the following search string was defined:
(“crowdsourcing experiment” OR “crowdsourcing application” OR “crowdsourcing platforms” OR “crowdsourcing case” OR “crowdsourcing tool” OR “plataforma de crowdsourcing”) AND (“urban planning” OR “urban design” OR “planejamento urbano”)

The searches were initially based on the Google Scholar and then performed at the ACM Digital Library, SpringerLink, Scopus, and ScienceDirect without time restriction. We also elaborated criteria for inclusion and exclusion of articles in order to select those studies that were relevant for answering the RQs.

### 3.3. Study Selection and Quality Assessment

The first step consisted of using the search string on the selected data sources, starting with Google Scholar as the primary search engine. At this stage, the only exclusion criterion was to exclude duplicate publications, resulting in a total of 412 articles. It is worth noting that some studies were added through backward snowball sampling, a technique used to complement the traditional systematic review process by identifying references that have cited the publications found in the search process [45]. The number of included and excluded studies during the selection process is shown in Figure 3.

Subsequently, in the second step, the analysis of titles and abstracts of the 412 articles was carried out. Inclusion criteria were as follows: (i) Be a complete, non-duplicate article; and (ii) the article should deal with an application, experiment, presentation, or framework of some crowdsourcing platform to support urban planning. Studies presenting non-peer reviewed contents (e.g., theses and dissertations) and editorials, summaries of conferences, extended abstracts, panel discussions, or introductions to special issues were removed. Furthermore, book chapters, tutorials, keynotes, and technical reports were also excluded and approximately 85 papers were selected.

In the third step, a brief read of 85 articles was carried out, with the inclusion criteria that the paper had the potential to answer the RQs. It is also noteworthy that the quality assessment was entirely conducted by the first author. The exclusion criteria remained the same as in step 2, since the mere reading of the title and abstract was sometimes not enough to see if there was a crowdsourcing platform applied to urban planning (or whether it was a conceptual or expository text on the subject, without a constructive approach). At the end of this step, 29 articles were selected.

The 29 articles were read completely in step 4. The inclusion criteria were applied based on a minimum quality score required. For the following questions presented in Table 1, the studies were evaluated as unresponsive, partially responding, and responding satisfactorily, being classified with 0, 0.5, or 1, respectively to assess the quality of the primary studies. The articles that scored higher than 50% of the possible points were chosen, resulting in 21 papers. Table A1 (Appendix A) provides a complete list of included studies.

After selecting the unique primary studies, the information about each venue in which the article was published was also recorded. Such metadata were stored for documentation and analysis purposes. The distribution of studies per type of venue is listed in Table A2 (Appendix B). The majority of the primary studies was published in journals (57%), but we also retrieved studies from conferences (33%) and workshops (9.5%). The venues with greater occurrences of primary studies were International Journal of Human-Computer Studies, ACM CHI Conference on Human Factors in Computing Systems, and IEEE International Conference on Pervasive Computing and Communication Workshops.

### 3.4. Data Extraction

A data extraction form was created to store all relevant data from primary studies. Only one author gathered metadata from the final set of articles taking into account each data extraction field. This process supported the entire classification and analysis procedures by matching each item to the corresponding value, as followed in previous work (e.g., [46]). The template used for data extraction is shown in Table 2.

### 3.5. Data Analysis and Synthesis

The first author extracted and analyzed the data, although this process involved all of the authors of this paper. In particular, the information for each item extracted was tabulated (see Section 4) and the resulting items were grouped to generate final results and to shed light on the RQs. Within each item, articles were analyzed and synthesized with respect to the crowdsourcing techniques being addressed or the specific crowdsourcing tasks used to support urban planning. Additional information on the review process (e.g., publication details) were recorded in the data collection form. Furthermore, disagreements were discussed with the principal author and resolved by consensus. This phase also included an examination of the results reported by articles addressing related topics, which were not directly included in the study selection process.

## 4. Results

### Overview

In this section, we analyze the results obtained from the synthesis of primary studies. In this sense, we provide an overview of the several aspects that current studies address in order to answer the RQs. The study selection resulted in 21 primary studies, published between 2011 and 2018. As it can be observed on the number of papers published per year, the occurrence of primary studies is low (i.e., only four papers) before the year 2015. This indicates that the interest on crowdsourcing for urban planning has been increasing in recent years, particularly in response to the need for solving complex urban problems affecting millions of people in different cities around the world.

Looking at the results provided by the selected articles, a total of 18 different crowdsourcing platforms were identified and analyzed using feature analysis [47], a qualitative method that is used in software engineering for assessing the relevance and implementation of features by candidate platforms and tools. Moreover, the information to answer the RQs was collected. However, before investigating the RQs in more depth, a classification of the platforms was made based on the auxiliary questions reported in the previous section. In particular, the selected platforms were grouped according to the generic crowdsourcing techniques: Tournament crowdsourcing (TC), open collaboration (OC), and virtual labor market (VLM), depending on how each article classifies its own approach into the following criteria: Competition of ideas, co-creation, participatory sensing, mobile crowdsourcing, folksonomy, and volunteered geographic information (VGI).

It is important to highlight the fact that most platforms were classified as OC, with more emphasis on VGI, mobile crowdsourcing, and participatory sensing. Six platforms explore TC-based approaches, and four of them incorporate open collaboration mechanisms such as comment and vote on proposed ideas [48]. Only one crowdsourcing approach uses a VLM platform [49]. The urban planning contexts addressed were also very diverse, highlighting the repetition of themes related to urban mobility, mapping of urban problems, and perception of landscapes. As previously noted, selected articles were published mostly in the 2015–2018 period (71.4%), which reflects the issue of topicality. Table 3 presents a summary of the most commonly addressed types of approaches found. It is likely that different platforms will concentrate on one or more of these techniques to differing degrees.


*(RQ1.1) Task definition: What kind of tasks were requested from participants?*


All the different tasks observed along the 18 selected platforms can be classified into the following categories: (i) Submitting ideas or projects; (ii) criticizing ideas or projects; (iii) voting; (iv) manually entering information into maps; (v) analyzing or classifying images; (vi) evaluating or classifying texts; (vii) photographic record; and (viii) answering specific questions.

As shown in Table 4, there is a direct relationship between the types of tasks requested and the crowdsourcing technique applied. For example, in ideas competition, there is a higher incidence of tasks i, ii, and iii. In addition, participatory sensing platforms denote a higher incidence of tasks v and viii. Those platforms classified as VGI basically incorporate task iv, while mobile crowdsourcing applications present a more diverse spectrum of tasks.


*(RQ1.2) User Interface (UI): How did the participants choose their tasks on interface?*


In the selected studies, three ways of making the tasks available in the UI were identified: (i) When the crowdsourcing process is exclusively composed of a single task, so in the UI, only one task is made available during the whole period in which the process is open; (ii) crowdsourcing is composed of more than one task, but only one task per step is available; and (iii) when multiple tasks are simultaneously available in the UI. Some aspects can be noticed from Table 5, including the fact that participatory sensing and VGI platforms do not provide multiple tasks on the interface simultaneously. Furthermore, ideas competition platforms make multiple interface tasks available simultaneously, in so far as they incorporate open collaboration mechanisms.


*(RQ1.3) Granularity: Do the tasks performed show traces of complexity, such as interdependence and coordination?*


Coordination is about managing the dependencies between tasks [14] and it can happen implicitly, from the structure itself, and explicitly through the communication between workers [50]. In this study, three levels of complexity were observed in task design, namely: (i) Lack of dependence between tasks and coordination; (ii) task dependency and implicit coordination; and (iii) task dependency and explicit coordination. As it can be seen in Table 6, around 61% of the selected platforms exhibit no coordination and no dependency between tasks, and all participatory sensing platforms are included in this group.


*(RQ1.4) Policy compensation: What kind of strategies and motivations for engagement can be identified?*


Three types of motivation have been identified, namely: (i) Civic or academic motives; (ii) awards; and (iii) fun and curiosity (Table 7). In five of them, the process was not explained. Of the 13 platforms in which motivational strategies involved are evident, it is clear that seven are based exclusively on civic or academic motives. Four platforms are based on fun and curiosity, where two of these are games with a purpose (GWAP). Overall, five platforms use prize incentives and only one of the 11 OC platforms identified involved some kind of reward strategies and motivations for engagement. 


*(RQ1.5) Work profile: Does the study report on assessments of the quality of work performed and the expertise (or reputation) of employees?*


In Table 8, a yes/no criteria was assigned to indicate whether requirements are established for (i) the reputation and expertise of workers, and (ii) to assess whether the quality of work performed has been reported. Of the 18 platforms, 12 had no entry barriers and any guest could participate. All the analyzed platforms reported on the quality of work processes.

Figure 4 summarizes the previous findings and answer the RQ1: “How does crowd work function in crowdsourcing platforms intended to support urban planning?”. According to our results, the crowdsourcing technique (e.g., ideas competition, mobile crowdsourcing, participatory sensing) is critical to understand the patterns along the five dimensions investigated.

Ideas competition. All platforms support tasks related to submitting an idea or project. Our results also suggest that 80% of such platforms allow users to vote while 60% enables criticizing on projects and ideas. In addition, only 20% have tasks related to photographic record and answering specific questions. In 60% of the platforms analyzed, tasks were available to be executed simultaneously. Basically, the idea is that users can submit their own ideas in parallel, criticize, and vote on other projects for determined time. Furthermore, 40% of analyzed platforms have task dependency with explicit coordination, and it occurs mainly because users can criticize different ideas incorporated and/or (re)submitted by authors. Regarding the type of motivation, 40% involve prizes to engage users and all platforms report academic or civic motivation. Furthermore, 40% of the platforms analyzed in this study require reputation or expertise. It is also worth noting that all platforms provide support for verifying the quality of work performed by means of evaluation strategies.

In mobile crowdsourcing, there is no information about crowdsourcing platforms for urban planning able to support the analysis and classification of images and similar items. On the other hand, 40% of the platforms provide support for submitting and/or criticizing ideas or projects, entering information into maps, analyzing or classifying texts, and providing photographic records. Results indicate that there are a small number of platforms (20%) with voting mechanisms and features for answering questions with high levels of specificity. Our findings highlight a reduced number of platforms and tools supporting single tasks in mobile crowdsourcing when compared to VGI and participatory sensing. This is also a pattern in the case of the platforms that require domain expertise and user reputation.

Regarding the study of VGI and participatory sensing, there is an absence of platforms for supporting tasks such as voting, criticizing, and submitting ideas or projects. Such types of crowdsourcing are more suitable for single tasks. In addition, it is important to note that these platforms differ in terms of task dependency and coordination.

## 5. Discussion

### 5.1. How Crowdsourcing Techniques and Quality Control Dimensions Used in Urban Planning can Be Used to Support the Urban Emergency Management of Mining-Related Dam Disruption

In this section, results based on previous review work [16] are extended for a more specific domain: The urban emergency management. This is possible as urban planning is a broader context with a focus on spatial socio-technical solutions as well. In this sense, strategies for user engagement concerning the techniques and computational tools or models applied can be reused on different urban emergency scenarios. The following topics are examples of crowdsourcing techniques being used to solve such problems:

• VGI techniques and mapping for emergency plans

Geospatial data based on personal local knowledge are of particular importance in disaster situations [51]. In order to support mapping for emergency response, a VGI approach could be applied (see Figure 5). For example, GeoCONAVI system [52] makes use of VGI to collect and analyze large volumes of data on forest fire events from social media using spatio-temporal information based on the context in which the hazard is occurring. On a map, geographic points, lines, and polygons can be drawn collaboratively by citizens, representing dwellings, sirens, meeting points, escape routes, special buildings, and roads.

All the VGI process could be based in alternated and sequential cycles of mapping and image analysis. In the mapping cycle, participants would be able to represent relevant geographical objects while in the image analysis cycle, they can verify and validate those mapped points, lines, and polygons. As it can be seen in Table 9, a common task type in VGI relies on manually entering information into maps, and it can be combined with an image analysis or classification task as a quality control mechanism [53]. Each cycle will be composed of only one task type, either mapping or validating the mapped objects.

According to the RQ3, VGI platforms mainly show task dependency. In the particular context of VGI for mapping emergency plans, dependency between tasks occurs through the mapping–validating cycles while coordination happens in an implicit way.

Strategies to engage participants can be based on civic or academic motives if the idea of building a collaborative emergency map is accepted. Furthermore, the whole mapping process can explore fun and curiosity, if a gamification process is adopted. Both in mapping and validation cycles, some geographical objects may require a specific expertise or enough reputation on the part of participants.

In their extensive review article, Haworth and Bruce [51] argued that the role of VGI in the pre-disaster planning phases (i.e., prevention/mitigation and preparation) of disaster management has been partially overlooked in the literature and this requires further consideration with large implications for risk reduction. As a result, volunteers can contribute to identify local vulnerabilities before the occurrence of disasters and thus generate efficient planning and response strategies.

• Mobile GWAP participatory sensing to support escape route simulations 

Through mobile crowdsourcing techniques, opportunities for citizen participation are created and a pool of participants around the city can engage in tasks and generate data in a distributed manner. Ubiquitous devices such as tablets and smartphones allow people to take pictures, to answer questions, to vote, to draw everywhere, and facilitate to register facts and ideas [54]. A GWAP is another strategy that may be used to extract information from a human player, which should be fun while performing tasks that are difficult to computers [55]. Also, GWAP can be performed on mobile devices, so participants can play and generate data about the city and public spaces.

In support to escape route simulations, a mobile GWAP could be applied. On an app with a navigation mode, a map with the escape routes, the user location, and the encounter point as a goal are plotted. The system will represent existing road signs. The game works with a timer, the participants must follow the escape route, catch points, and find the encounter point as fast as possible. At the end, a survey is applied. The collected data are used to evaluate the participant performance and to support new simulations, which characterize it as participatory sensing as well.

In the first stage, within the game, participants could walk through the streets following the escape routes and points, which will characterize a “enter information into a map” task type (see Table 10 for a typology of task types). The mapped information is related to the performance of participants in escaping. In the second stage, through a survey, the task type would be “answer specific question”. According to the RQ1, the task type “enter information into a map” is used mainly in VGI platforms [56] and mobile crowdsourcing [54,57], but the former can add not only locations, but also other spatial information such as mobility and the associated contexts [58]. The task type “answer specific question” is mainly used in participatory sensing approaches [59,60].

Those two stages would not be concomitant, so only one task will be available in the user interface per stage. As it can be observed in RQ2, GWAP platforms [55,61] and participatory sensing [59,62] have only one task along the whole process, so the mobile crowdsourcing approach (see Figure 6) proposed to support escape simulations can be considered as GWAP in the first stage and as participatory sensing in the second. 

The dependency between tasks occurs because the survey is a logical finalization of the simulation process, with no coordination between them supported by the app. It does not remove the possibility of presential coordination, since those simulations are made in groups and people talk to each other. According to the RQ3, GWAP and participatory sensing mainly show a lack of dependency and coordination between tasks.

In RQ4, “civic motivations” is described as the main strategy to engage participants, and in this scenario of emergency management, it fits perfectly. However, the GWAP strategy may add an extra motivation through the possibility of “curiosity and fun” while participants are playing. No expertise and reputation would be required, because it will be an open simulation to prepare population and the quality of tasks performed will be analyzed since it is a participatory sensing approach.

• Tournament crowdsourcing for emergency shelter and settlements design 

TC approaches have often been used to generate solutions to architectural design and urban planning. It consists of a challenge in a competition format, in which participants submit ideas to design a building or a public space, for example. One or more winners can emerge. In addition, some TC approaches use open collaboration mechanisms in which participants can vote, comment, and criticize ideas generated by their peers.

As part of the disaster prevention strategy, the location definition and design plans of the emergency settlements (including shelters, infrastructure, community buildings, and public spaces) should be developed. In this sense, a TC could be run in which interdisciplinary teams would be tasked with submitting their ideas for review. 

During the whole TC process, participants would be able to submit and criticize ideas and vote in parallel (see Table 11 for a typology of TC tasks for emergency shelter and settlements design). These three task types are mostly seen in the TC approaches [63] as can be observed in RQ1. According to the RQ2, OC mechanisms are merged into the TC process and multiple tasks are available in the interface simultaneously [64].

Since criticizing ideas would be a kind of task allowed, the need for explicit coordination will be a consequence. The combination of multiple tasks and explicit coordination may allow a more collaborative process of design, where participants can both compete and share ideas and evolve their solutions constantly.

The emergency shelter and settlements design process depicted in Figure 7 is a highly technical and complex task which need architects, engineers, and other technicians working cooperatively to address specific solutions for distinct cities. Awards/prizes are thus the most suitable incentive to attract those professional teams. Furthermore, reputation and expertise are also key factors to attract and retain teams and participants.

### 5.2. Crowd Participation in Urban Planning and Urban Emergency Management 

As we have seen in the overview of the previous section, crowdsourcing platforms used in urban planning have a diverse set of specific urban contexts and many of them are similar to emergency management as a concept. However, in urban emergency management [65], all those platforms would be involved in a broader problem, which is better described in the following question: How do we prepare and engage the population for a possible disaster without spreading panic and misinformation?

In this sense, crowdsourcing approaches and real-time collaboration should be articulated to complement themselves in urban planning and urban emergency settings. However, this process implies cognitive, emotional, and time costs since information needs to be collected, organized, and interpreted. Furthermore, it is still necessary to make an effort of communication, coordination of activities, and compatibility of schedules among participants [11,66]. In particular, as pointed out by Haworth and Bruce [51], there is a critical need for “timely and reliable communication in all aspects of disaster management”. Despite these concerns, it has been noted that real-time information produced by volunteers and relief agencies at the location of a disaster can result in catastrophic consequences (e.g., misinformation) if not treated appropriately. Especially useful for affected populations, infrastructure-independent and rapidly deployable communication systems with a long range can help in mitigating possible failures and overloads in such scenarios [67]. However, it is worth noting that there are manifested limitations associated with the limited network lifetime and battery power of participant devices, which leads to actionable insights for institutions and infrastructure support providers. Additional challenges faced by emergency managers are related to data security and privacy, scalability, credibility, mobility, uncertainty, and vagueness that are inherent in this kind of rapidly changing environments [15,51]. Furthermore, the ability of interpreting “heterogeneous, location-dependent information of various sources and quality” with high levels of flexibility has been unanimously considered to be an important issue during disaster response [68]. In such settings, authorities and volunteers need to cooperate in a trustworthy way. Moreover, the dynamic nature of disaster events distinguishes urban emergency management from other domains due to the need to form, adapt, and resize quickly [69].

### 5.3. Main Challenges in Task Design

According to the RQ1.5, in analyzing whether or not the quality of work has been achieved, it is reasonable to expect that the main challenges and questions for quality will be found in task design, since all platforms pay attention to the work profile**.** In order to propose crowdsourcing approaches to support urban emergency management, some key questions related to the design of effective crowdsourcing approaches include, but are not limited to: (i)How to distribute tasks along the crowdsourcing process?(ii)Among specialists and the public in general, how to select the most suitable participants for specific tasks and how to engage them?(iii)How to weight and merge crowdsourcing results in a transparent way and how to evaluate the efficiency of this applied heuristic?

### 5.4. Humans as Sensors

Besides the problem-solving aspect, where crowds can generate ideas, discuss, and make decisions collaboratively, crowdsourcing approaches may also explore humans as sensors. This capability is more evident on approaches focused on mapping tasks, such as VGI or participatory sensing with geographical location. 

In social media-supported crowdsourcing, for example, a Twitter user may be considered to be a sensor and a tweet created by a user will be sensory information [64]. This social sensor may be very noisy compared to physical sensors (such as heat sensors, kinetic sensors, light sensors). Crowd-based sensors have a wide variety of distinctive features. In particular, some participants are mostly online, while others are rarely active. Sometimes, these virtual sensors may not be working because they are out of coverage [70].

## 6. Conclusions, Limitations, and Future Work

In previous work, we had conducted a comprehensive literature review on crowdsourcing platforms in the context of urban planning [16]. The goal was to present a broad characterization of crowd work management in this context, with a special look at issues of quality of work. In this work, we extended the results of our previous study to a more specific domain: Urban emergency management. In particular, we were interested in understanding how crowdsourcing techniques and quality control dimensions used in urban planning could be used to support urban emergency management, especially in the context of mining-related dam outages.

Two major contributions were reported in this paper: (i) A five-dimension typology of quality in crowdsourcing, which can be leveraged for optimizing urban planning and emergency management processes; and (ii) a comparison between crowdsourcing tools designed for urban planning and urban emergency management, which was explored in the previous section and summarized in the Figure 8 with the Strengths, Weaknesses, Opportunities and Threats (SWOT) of each approach. We concluded that most of the facilities of urban planning crowdsourcing platforms can be applied to urban emergency management, as long as attention is paid to some specificities that are highlighted in Figure 8.

The framework draws attention to the social, technical, institutional, and policy aspects that need to be considered to manage the crowd work across these key dimensions. Concerning the strengths of using crowdsourcing in urban planning and emergency management, the possibility of asynchronous and remote crowd participation is most suitable for citizens that feel uncomfortable to contribute in face-to-face meetings or have difficulties to attend specific places or schedules. Through crowd decision documentation, the discussions and results involved in the solution process are documented in the crowdsourcing system, which makes it more transparent and safe. Once citizens can investigate problems, submit ideas, or observations, they strengthen relationships with the cities and places around them. As a result, crowds can generate different solutions and perspectives for the same problem. 

Especially in the context of urban emergency management, the problems are more tactile, urgent, and the sense of relevance is bigger when compared to urban planning. If implemented successfully, crowdsourcing tools and strategies can be replicated in different cities with similar problems and contexts while contributing to massive crowd engagement. In urban emergency management, people from different places can identify with the relevance and urgency of the proposal and feel motivated to collaborate in the crowdsourcing process. However, using crowdsourcing platforms has some drawbacks. For instance, finding suitable workers and matching them according to the task requirements implies that participants feel motivated to engage. At the same time, the task execution often requires special skills or reputation. Moreover, the crowd can feel unmotivated and drop the process if the goals are not clearly stated. Another potential drawback is related to the exclusion of a part of the population that is uncomfortable to participate in such digital events or social media. This is even more remarkable for elderly people who can have very particular viewpoints and extensive knowledge about the city. In addition, one potential threat of using crowdsourcing in real-world scenarios is the external manipulation of the outcomes generated by the crowd. Furthermore, private information can be leaked and this is particularly dangerous in the context of urban emergency management.

Through a critical lens, our systematic review inevitably has certain limitations regarding the bias in the selection of primary studies. This situation was also reported in previous work (e.g., [70]). Thus, our review cannot be comprehensive with respect to the total number of potential secondary studies about crowdsourcing in urban planning and emergency response. Furthermore, there are also problems concerning the potential inaccuracies ascribed to the data extraction process. In particular, several publications lacked information about technology use in real-world settings. Another potential bias lies in the fact that only one of the authors decided whether or not to include papers. Even in a small way, we have diminished the inconsistencies related to the methods used to select, categorize, and synthesize findings from primary studies.

Predicated on using the resources of the crowd to address urban emergency management in highly dynamic scenarios, the results of this study contribute to the recognition of the current landscape by observing tendencies, limitations, and opportunities for research in this field, besides serving as a reference for the implementation of future crowdsourcing solutions. While this research is limited to the context studied in the primary studies, future research aims at understanding the use of social media in disaster prevention and urban planning and how heuristics or applied algorithms could be used to synthesize and generate results from crowd inputs.

## Figures and Tables

**Figure 1 sensors-19-05235-f001:**
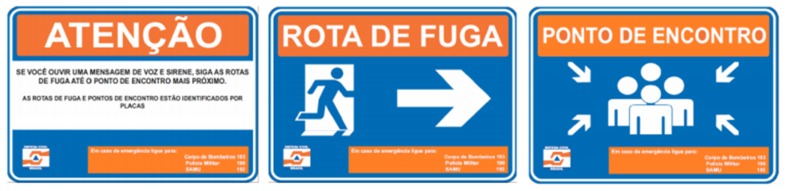
Emergency signposts warning the population about escape routes and meeting points [12].

**Figure 2 sensors-19-05235-f002:**
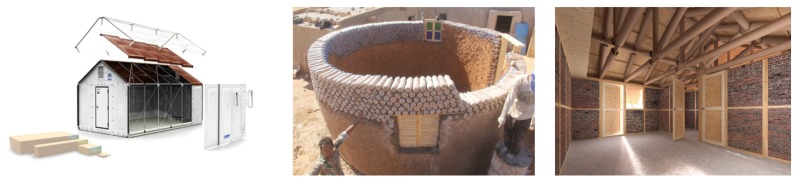
Immediate shelters vs. temporary vs. rebuilding [39].

**Figure 3 sensors-19-05235-f003:**
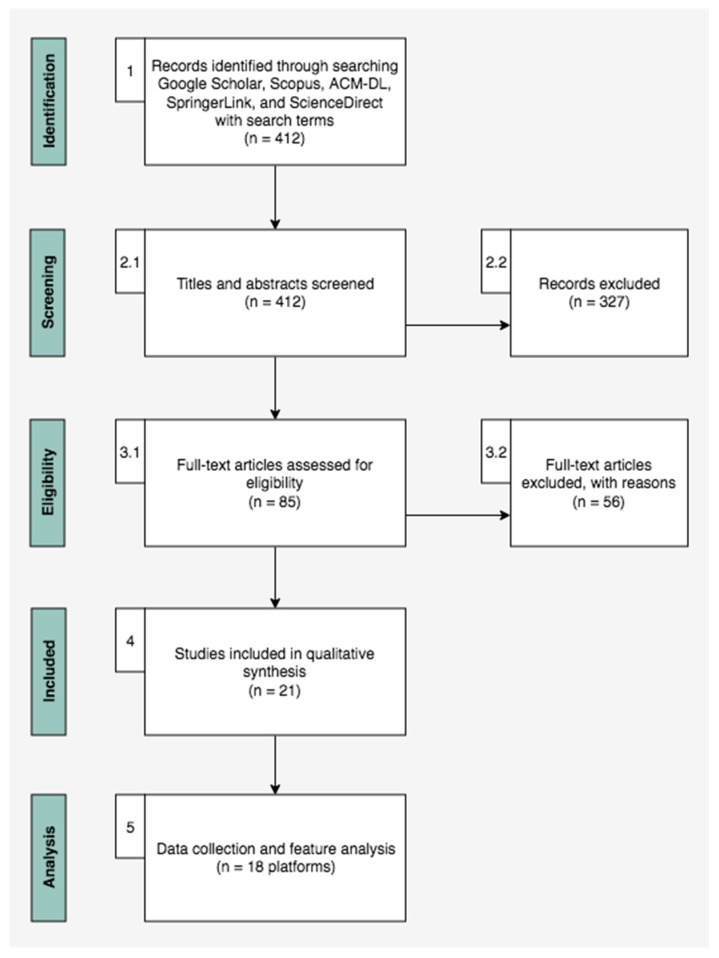
Flow diagram of searching, screening, inclusion, and analysis of publications.

**Figure 4 sensors-19-05235-f004:**
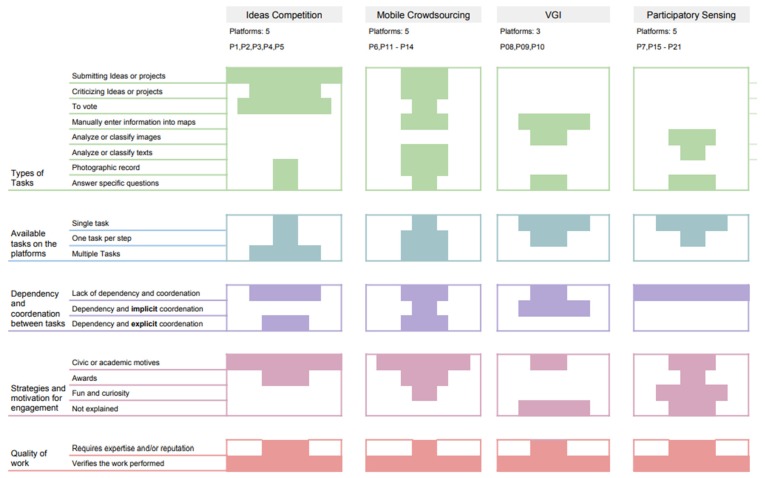
Five-dimension typology of quality in crowd work per crowdsourcing technique in urban planning.

**Figure 5 sensors-19-05235-f005:**
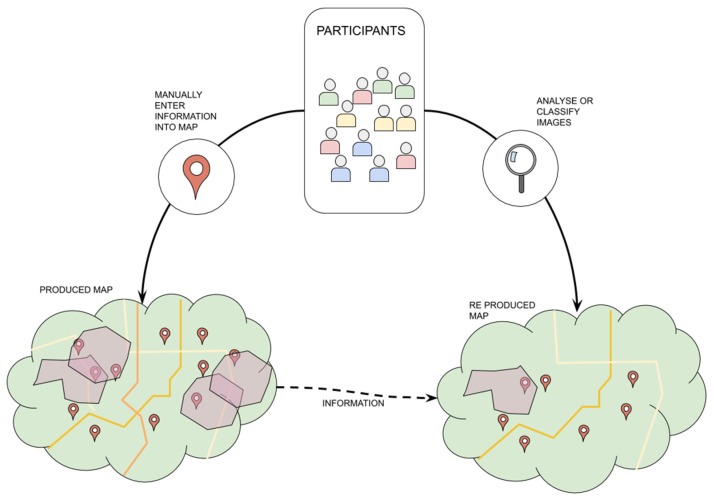
VGI approach for emergency plans.

**Figure 6 sensors-19-05235-f006:**
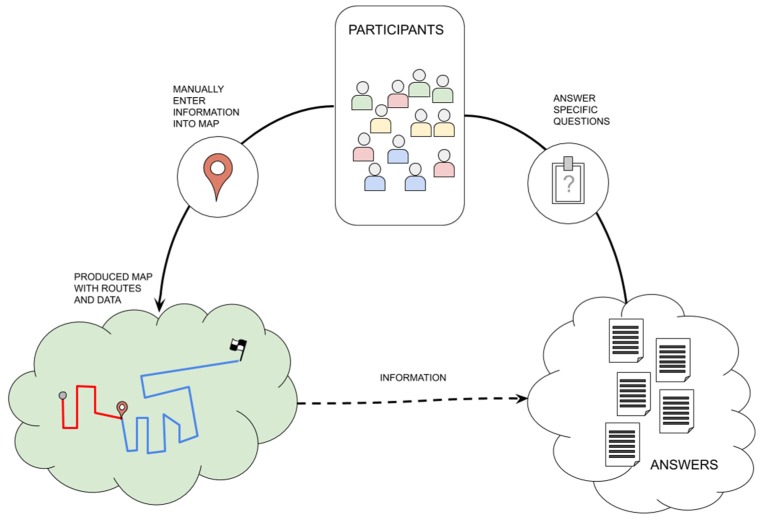
Mobile games with a purpose (GWAP) participatory sensing approach for escape route simulations.

**Figure 7 sensors-19-05235-f007:**
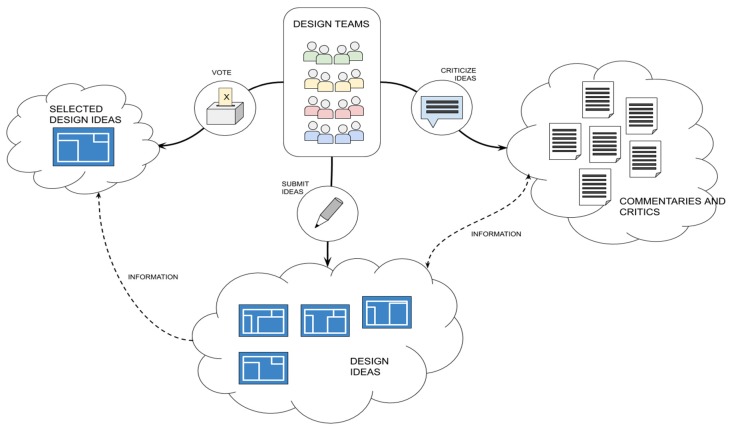
Tournament crowdsourcing approach for emergency shelter and settlements design.

**Figure 8 sensors-19-05235-f008:**
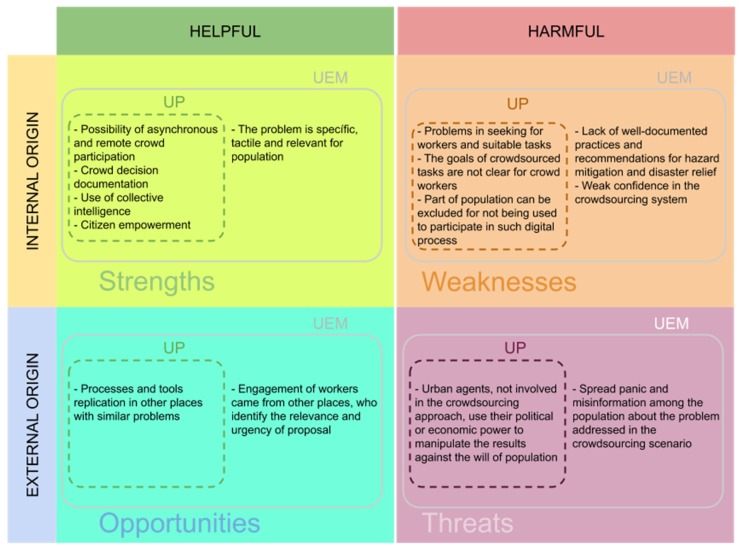
SWOT analysis for crowdsourcing approaches in urban planning and emergency management.

**Table 1 sensors-19-05235-t001:** Quality criteria.

Question	Score
Does the article indicate what kind of tasks the participants were asked to do?	Yes/Partially/No
Is there evidence demonstrating how the participants chose their tasks?	Yes/Partially/No
Does the article indicate if the tasks performed show traces of complexity?	Yes/Partially/No
Is the study revealing the strategies and motivations for engagement?	Yes/Partially/No
Does the article report on assessments of the quality of work performed and the reputation of workers?	Yes/Partially/No

**Table 2 sensors-19-05235-t002:** Data extraction form.

Data Item	Description
Study ID	Unique identifier for the article
Year	Year of publication
Title	Title of the article
Author(s)	Name of the author(s)
Keywords	List of keywords used to represent the content of the paper
Study selection	Inclusion/exclusion choice and quality evaluation score
Crowdsourcing technique(s)	Type of techniques used for data collection and production in crowdsourcing
Platform	Crowdsourcing platform(s) used for gathering data in urban planning scenarios
Urban planning context	Describes the context and environment within which the crowdsourcing activity occurs
Task(s)	Type(s) of tasks supported by crowdsourcing platforms
User Interface (UI)	How tasks are made available in UI along the crowdsourcing process
Dependency and coordination	Degree of coordination and task dependency
Motivation	Strategies used to motivate crowd participants
Worker profile	Mechanisms used for assessing quality in crowdsourcing, such as expertise and reputation

**Table 3 sensors-19-05235-t003:** Classification of platforms according to crowdsourcing techniques.

Crowdsourcing Techniques		Platform	Year	Urban Planning Specific Context	Ref.
**Tournament crowdsourcing**	Ideas competition	Future Profiling Schools	2013	Architecture competition	P1
		Participatory Budgeting	2018	Participatory budgeting	P2
		NextStop Design	2012	Urban mobility	P3
**Tournament crowdsourcing/Open collaboration**	Ideas competition w/ co-creation				
		YouCity Challenge	2017	Urban mobility	P4
	Ideas competition/participatory sensing	CommunityCrit	2018	Urban design	P5
	Mobile crowdsourcing/GWAP	PhotoCity	2011	Architecture tridimensional modeling	P6
**Open collaboration**	Folksonomy/GWAP	Sus-tweet-ability	2014	Urban infrastructure	P7
	Volunteered (or crowdsourced) Geographical Information – VGI	Austin Historical	2015	Architectural heritage	P8
		PEDS	2017	Urban mobility	P9
		Open Street Map	2016	Humanitarian mapping	P10
	Mobile crowdsourcing	SenseCityVity	2017	Urban problems mapping	P11
		Context Weaver	2015	Urban problems mapping and discussion	P12
		Community Reminder	2016	Urban security	P13
		Community Circles	2015	Urban problems mapping and discussion	P14
	Participatory sensing	Atmos	2016	Wheather	P15
		mPASS	2015	Urban mobility	P16
			2014		P17
		Como é Campina?	2017	Urban landscape	P18
**Virtual labor market**		Mechanical Turk	2015	Landscape and architectural perception	P19
			2015		P20
			2017		P21

**Table 4 sensors-19-05235-t004:** Types of tasks addressed on selected platforms.

Crowdsourcing Techniques		Platform	Types of Tasks Addressed on Selected Platforms							
			Submitting Ideas or Projects	Criticizing Ideas or Projects	To Vote	Manually Enter Information into Maps	Analyze or Classify Images	Evaluate or Classify Texts	Photographic Record	Answer Specific Questions
**Tournament crowdsourcing**	Ideas competition	Future Profiling Schools	X							
		Participatory Budgeting	X		X					
		NextStop Design	X	X	X					
**Tournament crowdsourcing/Open collaboration**	Ideas competition w/ co-creation									
		YouCity Challenge	X	X	X					
	Ideas competition/ participatory sensing	CommunityCrit	X	X	X				X	X
	Mobile crowdsourcing/GWAP	PhotoCity							X	
**Open collaboration**	Folksonomy/ GWAP	Sus-tweet-ability						X		
	Volunteered (or crowdsourced) Geographical Information	Austin Historical Survey Wiki				X				
		PEDS								X
		Open Street Map				X	X			
	Mobile crowdsourcing	SenseCityVity	X			X	X	X	X	
		Context Weaver		X		X				
		Community Reminder							X	X
		Community Circles	X	X	X					
	Participatory sensing	Atmos								X
		mPASS								X
		Como é Campina?					X			X
**Virtual labor market**		Mechanical Turk					X			

**Table 5 sensors-19-05235-t005:** Available tasks on the platforms.

Crowdsourcing Techniques		Platform	Available Tasks on the Platforms		
			A Single Task during the Whole Process	Only One Task Per Step Is Available	Multiple Tasks Are Simultaneously Available in the UI
**Tournament crowdsourcing**	Ideas competition	Future Profiling Schools	X		
		Participatory Budgeting		X	
		NextStop Design			X
**Tournament crowdsourcing/Open collaboration**	Ideas competition w/ co-creation				
		YouCity Challenge			X
	Ideas competition/ participatory sensing	CommunityCrit			X
	Mobile crowdsourcing/GWAP	PhotoCity	X		
**Open collaboration**	Folksonomy/GWAP	Sus-tweet-ability	X		
	Volunteered (or crowdsourced) Geographical Information	Austin Historical Survey Wiki	X		
		PEDS	X		
		Open Street Map		X	
	Mobile crowdsourcing	SenseCityVity			X
		Context Weaver			X
		Community Reminder		X	
		Community Circles			X
	Participatory sensing	Atmos	X		
		mPASS	X		
		Como é Campina?		X	
**Virtual labor market**		Mechanical Turk	X		

**Table 6 sensors-19-05235-t006:** Dependency and coordination between tasks.

Crowdsourcing Techniques		Platform	Dependency and Coordination between Tasks		
			Lack of Dependency between Tasks and Coordination	Task Dependency and Implicit Coordination	Task Dependency and Explicit Coordination
**Tournament crowdsourcing**	Ideas competition	Future Profiling Schools	X		
		Participatory Budgeting	X		
		NextStop Design	X		
**Tournament crowdsourcing/Open collaboration**	Ideas competition w/ co-creation				
		YouCity Challenge			X
	Ideas competition/ participatory sensing	CommunityCrit			X
	Mobile crowdsourcing/GWAP	PhotoCity	X		
**Open collaboration**	Folksonomy/GWAP	Sus-tweet-ability	X		
	Volunteered (or crowdsourced) Geographical Information	Austin Historical Survey Wiki		X	
		PEDS	X		
		Open Street Map		X	
	Mobile crowdsourcing	SenseCityVity			X
		Context Weaver			X
		Community Reminder	X		
		Community Circles		X	
	Participatory sensing	Atmos	X		
		mPASS	X		
		Como é Campina?	X		
**Virtual labor market**		Mechanical Turk	X		

**Table 7 sensors-19-05235-t007:** Strategies and motivations for engagement.

Crowdsourcing Techniques		Platform	Strategies and Motivations for Engagement			
			Civic or Academic Motives	Awards	Fun and Curiosity	Not Explained
**Tournament crowdsourcing**	Ideas competition	Future Profiling Schools	X	X		
		Participatory Budgeting	X			
		NextStop Design	X			
**Tournament crowdsourcing/Open collaboration**	Ideas competition w/ co-creation					
		YouCity Challenge	X	X		
	Ideas competition/ participatory sensing	CommunityCrit	X			
	Mobile crowdsourcing/GWAP	PhotoCity	X	X	X	
**Open collaboration**	Folksonomy/GWAP	Sus-tweet-ability	X		X	
	Volunteered (or crowdsourced) Geographical Information	Austin Historical Survey Wiki				X
		PEDS				X
		Open Street Map	X			
	Mobile crowdsourcing	SenseCityVity	X			
		Context Weaver	X	X		
		Community Reminder				X
		Community Circles	X			
	Participatory sensing	Atmos			X	X
		mPASS				X
		Como é Campina?			X	
**Virtual labor market**		Mechanical Turk	X	X		

**Table 8 sensors-19-05235-t008:** Assessments of the quality of work performed and the expertise (or reputation) of employees.

Crowdsourcing Techniques		Platform	Requirements Are Established for the Reputation and Expertise of Workers		To Assess whether the Quality of Work Performed has Been Reported	
			No	Yes	No	Yes
**Tournament crowdsourcing**	Ideas competition	Future Profiling Schools		X		X
		Participatory Budgeting	X			X
		NextStop Design	X			X
**Tournament crowdsourcing/Open collaboration**	Ideas competition w/ co-creation					
		YouCity Challenge		X		X
	Ideas competition/ participatory sensing	CommunityCrit	X			X
	Mobile crowdsourcing/GWAP	PhotoCity	X			X
**Open collaboration**	Folksonomy/GWAP	Sus-tweet-ability	X			X
	Volunteered (or crowdsourced) Geographical Information	Austin Historical Survey Wiki	X			X
		PEDS		X		X
		Open Street Map	X			X
	Mobile crowdsourcing	SenseCityVity	X			X
		Context Weaver	X			X
		Community Reminder	X			X
		Community Circles		X		X
	Participatory sensing	Atmos	X			X
		mPASS	X			X
		Como é Campina?		X		X
**Virtual labor market**		Mechanical Turk		X		X

**Table 9 sensors-19-05235-t009:** A typology of volunteered geographic information (VGI) task types for emergency plans.

Task Type	Available Task in UI	Dependency and Coordination between Tasks	Strategies and Motivation for Engagement	Assessments of the Quality of Work Performed/ Expertise
Manually enter information into maps Analyze or classify images	Only one task per stage	Task dependency and implicit coordination	Civic or academic motives	Assessment of the quality of work
			Fun and curiosity	Expertise or reputation required

**Table 10 sensors-19-05235-t010:** A typology of participatory sensing tasks for escape route simulations.

Task Type	Available Task in UI	Dependency and Coordination between Tasks	Strategies and Motivation for Engagement	Assessments of the Quality of Work Performed/Expertise
Manually enter information into maps	Only one task per stage	Lack of dependency between tasks and coordination	Civic or academic motives	Assessment of the quality of work
Answer specific questions			Fun and curiosity	No expertise or reputation required

**Table 11 sensors-19-05235-t011:** A typology of tournament crowdsourcing (TC) tasks for emergency shelter and settlements design.

Task Type	Available Task in UI	Dependency and Coordination between Tasks	Strategies and Motivation for Engagement	Assessments of the Quality of work Performed and the Expertise
Submit ideas Criticize ideas Vote	Multiple tasks	Task dependency and explicit coordination	Awards	Assessment of the quality of work Expertise or reputation required

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
