# Peer review of "Crowdsourcing as a Tool for Urban Emergency Management: Lessons from the Literature and Typology"

_sensors, 2019, doi:10.3390/s19235235_

Round 1

Reviewer 1 Report

The title of the paper is promising but in reality it is probably exaggerated taking into account the content of the study. As I understand, the purpose of the study was to answer the question "how can CS techniques and quality control dimensions used in urban planning be used to support the management of mining-related dam disruption emergencies?" The conclusions make it difficult to find an unambiguous answer to this question - in my opinion such an answer should be clearly stated in the “conclusion” section..

In the context of the topic of the paper, I do not understand why Authors focused on the remaining 6 research questions, included the main of which is "to support urban planning" by crowdsourcing platforms. Support urban planning seems to be a bit more general issue while the title of the paper points to "Urban Emergency Management" - emergency management is a much more detailed issue than "urban planning", am I wrong ? 

In my opinion, the issue of the usefulness of crowdsourcing to "emergency management" should be definitely more emphasized in the research section and in the conclusions.

 Taking into account a large number of questions (RQ, RQ1-RQ6), a graphical form of the "conceptual framework" could be useful. It would illustrate the relationships between specified research elements and demonstrate their relationship to the key research question, i.e. questions about the use of crowdsourcing for "emergency management" (specifically in cases of mining-related dam rupture emergencies). 

To make the assessment of the usefulness of "crowdsourcing" to "emergency management" more complete, it would be useful to prepare a table in which the strengths and weaknesses of such a solution would be demonstrated. 

In general, I positively assess the applied research methodology, but I am not convinced if delivering such details as the way of discussions (WhatasApp, via-email) is relevant in the context of the discussed research problem. 

In my opinion, if the topic is about "emergency management" it is worth considering the reference to the theory of crisis management (at the moment there is only reference to quality management).

Author Response

Dear Reviewer,

We would like to express our sincere gratitude for the opportunity to submit a revised version of our manuscript. We also understand the rich and diverse comments and improvement suggestions and merged all the valuable contribution in the (major) revised version that we are submitting.

The paper was substantially and carefully revised, and a lot of changes have taken place. We have sent the revised manuscript and answered in detail to each comment. At the following, the points mentioned by the reviewers are discussed, and the answers are structured by the degree of similarity between reviewers’ comments for readability.

Reviewer:

In general, I positively assess the applied research methodology, but I am not convinced if delivering such details as the way of discussions (WhatsApp, via-email) is relevant in the context of the discussed research problem.

A: We appreciate the positive feedback from the reviewer. We removed the text about using WhatsApp, email and Google Docs as discussion channels. The general text of the subsection was also rewritten to maintain consistency. As a result, the details about the methodological process were refined.

Reviewer:

The title of the paper is promising but in reality it is probably exaggerated taking into account the content of the study. As I understand, the purpose of the study was to answer the question "how can CS techniques and quality control dimensions used in urban planning be used to support the management of mining-related dam disruption emergencies?" The conclusions make it difficult to find an unambiguous answer to this question - in my opinion such an answer should be clearly stated in the “conclusion” section.

A: We agree with the comments and have focus our improvement efforts to reframe the way as the article is structured. With regards to the title of the paper, it was slightly changed to “Crowdsourcing as a Tool for Urban Emergency Management: Lessons from the Literature and Typology” in order to reflect the five-dimension typology of quality in crowdsourcing that was added in this revised version (Figure 3). It is also worth noting that the abstract was also carefully revised and the new organization of contents aims to deliver a clear perspective of the relation between the papers in the literature review and the way they contribute to the proposed conceptual framework. The discussion section was reformulated with more focus on the comparison between the results obtained in the previous study (i.e., [16]) and the findings obtained in this review in order to extend to the context of urban emergency management, especially in the context of mining-related dam disasters.

Reviewer:

In my opinion, the issue of the usefulness of crowdsourcing to “emergency management” should be definitely more emphasized in the research section and in the conclusions.

Taking into account a large number of questions (RQ, RQ1-RQ6), a graphical form of the “conceptual framework” could be useful. It would illustrate the relationships between specified research elements and demonstrate their relationship to the key research question, i.e. questions about the use of crowdsourcing for “emergency management” (specifically in cases of mining-related dam rupture emergencies).

In the context of the topic of the paper, I do not understand why Authors focused on the remaining 6 research questions, included the main of which is "to support urban planning" by crowdsourcing platforms. Support urban planning seems to be a bit more general issue while the title of the paper points to "Urban Emergency Management" - emergency management is a much more detailed issue than "urban planning", am I wrong ?It is not clear in the paper of the relationships between the review results of crowdsourcing and urban emergency management in section 4. Moreover, the authors just describe how the review results (RQ1-RQ5) apply in urban emergency management simply.

A: Thank you for pointing this out. We agree with this and have incorporated your suggestion throughout the manuscript. Urban planning is a broader context with a focus on spatial socio-technical solutions as well. According to the suggestion, we revised the content of the Results and Discussion sections in order to explain the different techniques and models used in crowdsourcing for emergency management. In particular, Figures 4-6 characterize different crowdsourcing approaches for emergency management scenarios (e.g., VGI, Mobile GWAP, Tournament Crowdsourcing). We have also reformulated the RQs in the Methodological section in order to subdivide the main research question (RQ1) into five subquestions (RQ1.1-RQ1.5). The answers to such questions are provided in the Results section and a graphical form of the conceptual framework was added.

Reviewer:

To make the assessment of the usefulness of “crowdsourcing” to “emergency management” more complete, it would be useful to prepare a table in which the strengths and weaknesses of such a solution would be demonstrated.

A: As suggested by the reviewer, we have added a SWOT analysis for crowdsourcing approaches in urban planning and emergency management (Figure 7).

Reviewer:

In my opinion, if the topic is about "emergency management" it is worth considering the reference to the theory of crisis management (at the moment there is only reference to quality management).

A: We agree with the comment of the reviewer and a vast set of lessons from the literature were incorporated in the revised version. This includes studies on crowdsourcing for emergency management and disaster response assessment [61, 65-66], social media data analysis of urban emergency events [62], barriers in crisis management [63], and communication challenges in post-disaster scenarios [64]. In addition, we have also explored some of the current challenges of VGI for disaster management [48]. Further examples of crowdsourcing systems were added (e.g., CrowdHydrology, Social.Water, and GeoCONAVI).

Thanks for the opportunity to major revise the manuscript. We expect to fulfil your comments and suggestions.

With kind regards,

The manuscript authors,

Ramon Chaves

Daniel Schneider

António Correia

Claudia Motta

Marcos Borges

Reviewer 2 Report

The authors intent to extract some lessons and insights from current crowdsourcing literatures to address several important problems in urban emergency management field. It's an interesting work and structured well. But there are some problems the authors would consider while improving the paper. 1. Why the authors just take into account literatures of crowdsourcing for urban planning or ubran design while neglect other crowdsourcing models, such as crowdsourcing for idea innovation, for problem solving, for other complex or simple tasks. A result of considering the crowdsourcing for ubran management solely is the limitation of number of literatures. It, to some degree, can not convince readers greatly that the lessons or strategies learned from these literatures with so limited literatures. There are some measures for quality control in other crowdsourcing articles can help to develop lessons for urban emergency management. for example, the quality control measurements mentioned in "Daniel F, Kucherbaev P, Cappiello C, et al. Quality control in crowdsourcing: A survey of quality attributes, assessment techniques, and assurance actions. ACM Computing Surveys, 2018. 2. The paper presents a little content related to urban emergency management and current studies related to this field. Further, the definition of crowdsourcing and its current studies are missed as well. 3. It is not clear in the paper of the relationships between the reivew results of crowdsourcing and urban emergency management in section 4. Moreover, the authors just describe how the review results (RQ1-RQ5) apply in urban emergency management simply. 5. The section of discussion and future direction is too simple and does not connect with the theme of this paper tightly. By the way, the limitation part is missed. 6. I guess, the number of references cited in the paper is not in right order.

Author Response

Dear Reviewer,

We would like to express our sincere gratitude for the opportunity to submit a revised version of our manuscript. We also understand the rich and diverse comments and improvement suggestions and merged all the valuable contribution in the (major) revised version that we are submitting.

The paper was substantially and carefully revised, and a lot of changes have taken place. We have sent the revised manuscript and answered in detail to each comment. At the following, the points mentioned by the reviewers are discussed, and the answers are structured by the degree of similarity between reviewers’ comments for readability.

Reviewer:

The paper presents a little content related to urban emergency management and current studies related to this field. Further, the definition of crowdsourcing and its current studies are missed as well.

Why the authors just take into account literatures of crowdsourcing for urban planning or urban design while neglect other crowdsourcing models, such as crowdsourcing for idea innovation, for problem solving, for other complex or simple tasks. A result of considering the crowdsourcing for urban management solely is the limitation of number of literatures. It, to some degree, can not convince readers greatly that the lessons or strategies learned from these literatures with so limited literatures. There are some measures for quality control in other crowdsourcing articles can help to develop lessons for urban emergency management. for example, the quality control measurements mentioned in "Daniel F, Kucherbaev P, Cappiello C, et al. Quality control in crowdsourcing: A survey of quality attributes, assessment techniques, and assurance actions. ACM Computing Surveys, 2018.

A: We thank the reviewer’s suggestion. With regards to the literature review, we agree with the comment of the reviewer and a vast set of lessons from the literature were incorporated in the revised version. This includes studies on crowdsourcing for emergency management and disaster response assessment (e.g., [61, 65-66]), social media data analysis of urban emergency events (e.g., [62]), barriers in crisis management (e.g., [63]), and communication challenges in post-disaster scenarios (e.g., [64]). In addition, we have also explored some of the current challenges of VGI for disaster management (e.g., [48]). Further examples of crowdsourcing systems were added to this revised version, including CrowdHydrology [32], Social.Water [33], and GeoCONAVI [49]. It is also worth noting that the “quality control” dimension was further explored through the reference suggested by the reviewer (i.e., [19]). As the major contribution of the paper is the conceptual framework proposed, the presentation and discussion of the challenges and gaps in the current state of the art were reinforced based on new literature studies carefully reviewed.

Reviewer:

I guess, the number of references cited in the paper is not in right order.

A: The suggested correction has been made and the reference list has been updated and renumbered.

Reviewer:

It is not clear in the paper of the relationships between the review results of crowdsourcing and urban emergency management in section 4. Moreover, the authors just describe how the review results (RQ1-RQ5) apply in urban emergency management simply.

The section of discussion and future direction is too simple and does not connect with the theme of this paper tightly. By the way, the limitation part is missed.

A: We appreciate the positive feedback from the reviewer. A substantial revision was made and the Discussion section was reformulated in order to answer the following question: “How crowdsourcing techniques and quality control dimensions used in urban planning can be used to support the urban emergency management of mining-related dam disruption?”. In this sense, we put more focus on the comparison between the results obtained in the previous study (i.e., [16]) and the findings obtained in this review in order to extend to the context of urban emergency management, especially in the context of mining-related dam disasters. According to the suggestion, we revised the content of the Results and Discussion sections in order to explain the different techniques and models used in crowdsourcing for emergency management. In particular, Figures 4-6 characterize different crowdsourcing approaches (e.g., VGI, Mobile GWAP, Tournament Crowdsourcing) for emergency management scenarios. We have also reformulated the RQs in the Methodological section in order to subdivide the main research question (RQ1) into five subquestions (RQ1.1-RQ1.5). The answers to such questions are provided in the Results section. In addition, we have added a five-dimension typology of quality in crowdsourcing (Figure 3). This framework presents a graphical overview of the results achieved from our systematic literature review.

Furthermore, as suggested by the reviewer, we have improved Section 6 in order to make the assessment of the usefulness of crowdsourcing to emergency management more complete. To do this, we have added a SWOT analysis for crowdsourcing approaches in urban planning and emergency management (Figure 7) and further expanded the conclusions of our work.

Thanks for the opportunity to major revise the manuscript. We expect to fulfil your comments and suggestions.

With kind regards,

The manuscript authors,

Ramon Chaves

Daniel Schneider

António Correia

Claudia Motta

Marcos Borges

Reviewer 3 Report

The forms for Researh Questions (RQ1-RQ6) from page no. 5 of the article are not appropriate forms. Normally, a Research Question should question the null hypothesis, but in this article they were not formulated in this style. Each Research Question (RQ) is completed with a simple centralizing table (pages 8-23).

I would suggest modifying RQ1-RQ6 in the form of testing null hypotheses and then applying validity tests to confirm / inform each hypothesis based on statistical data.

From my point of view, the article is just a summary of the literature review.

Author Response

Dear Reviewer,

We would like to express our sincere gratitude for the opportunity to submit a revised version of our manuscript. We also understand the rich and diverse comments and improvement suggestions and merged all the valuable contribution in the (major) revised version that we are submitting.

The paper was substantially and carefully revised, and a lot of changes have taken place. We have sent the revised manuscript and answered in detail to each comment. At the following, the points mentioned by the reviewers are discussed, and the answers are structured by the degree of similarity between reviewers’ comments for readability.

Reviewer:

I would suggest modifying RQ1-RQ6 in the form of testing null hypotheses and then applying validity tests to confirm / inform each hypothesis based on statistical data.

A: We appreciate the positive feedback from the reviewer. According to the suggestion, we have reformulated the RQs in the Methodological section in order to subdivide the main research question (RQ1) into five subquestions (RQ1.1-RQ1.5). The answers to such questions are provided in the Results section and a five-dimension typology of quality in crowdsourcing (Figure 3) was added in this revised version. This framework presents a graphical overview of the results achieved from our systematic literature review and allowed to characterize each finding based on statistical data. In addition, we have also added a SWOT analysis for crowdsourcing approaches in urban planning and emergency management (Figure 7) and further expanded the conclusions of our work.

Thanks for the opportunity to major revise the manuscript. We expect to fulfil your comments and suggestions.

With kind regards,

The manuscript authors,

Ramon Chaves

Daniel Schneider

António Correia

Claudia Motta

Marcos Borges

Reviewer 4 Report

The authors conducted a systematic and comprehensive review of literature on the topic of "crowdsourcing as a tool for urban emergency management. " The reviewer does not have specific suggestions/comments to improve the manuscript. 

Author Response

Dear Reviewer,

We would like to express our sincere gratitude for the opportunity to submit a revised version of our manuscript. The paper was substantially and carefully revised, and a lot of changes have taken place. We have sent the revised manuscript.

Thanks for the opportunity to major revise the manuscript. We expect to fulfil your comments and suggestions.

With kind regards,

The manuscript authors,

Ramon Chaves

Daniel Schneider

António Correia

Claudia Motta

Marcos Borges

Round 2

Reviewer 2 Report

The paper has been improved greatly. further, with respect to measures of quality control in crowdsourcing, the following recent articles can be cited and considered. 1. Lukyanenko R, Wiggins A, Rosser H K. Citizen Science: An Information Quality Research Frontier,Information Systems Frontiers, 2019.
2. Zhang X, Su J. A combined fuzzy DEMATEL and TOPSIS approach for estimating participants in knowledge-intensive crowdsourcing,Computers & Industrial Engineering, 2019, 137.
3. Gong, Y.. Estimating participants for knowledge-intensive tasks in a network of crowdsourcing marketplaces.  Information Systems Frontiers, 2017.

Author Response

Dear Reviewer,

We would like to express our sincere gratitude for the opportunity to submit a revised version of our manuscript. We also understand the improvement suggestions and merged all the valuable contribution in the (minor) revised version that we are submitting.

The paper was carefully revised and a few changes have taken place. We have sent the revised manuscript. At the following, the points mentioned by the reviewer are discussed.

Reviewer:

The paper has been improved greatly. Further, with respect to measures of quality control in crowdsourcing, the following recent articles can be cited and considered.

Lukyanenko R, Wiggins A, Rosser H K. Citizen Science: An Information Quality Research Frontier,Information Systems Frontiers, 2019. Zhang X, Su J. A combined fuzzy DEMATEL and TOPSIS approach for estimating participants in knowledge-intensive crowdsourcing. Computers & Industrial Engineering, 2019, 137. Gong, Y.. Estimating participants for knowledge-intensive tasks in a network of crowdsourcing marketplaces.  Information Systems Frontiers, 2017.

Authors:

We appreciate the positive feedback from the reviewer. According to the suggestion, we revised the content of subsection 2.1 in order to incorporate the challenging issues in quality control as reported in the studies mentioned by the reviewer. This has brought considerable value to the literature review. Furthermore, it is also worth noting that the Conclusion section was carefully revised in order to explain the SWOT analysis (Figure 7) and various other improvements were made (e.g., English writing).

Thanks for the opportunity to revise the manuscript. We expect to fulfil your comments and suggestions.

With kind regards,

The manuscript authors,

Ramon Chaves
Daniel Schneider
António Correia
Claudia Motta
Marcos Borges

Reviewer 3 Report

The article is better then the first version.

Author Response

Dear Reviewer,

We would like to express our sincere gratitude for the opportunity to submit a revised version of our manuscript. We also understand the improvement suggestions and merged all the valuable contribution in the (minor) revised version that we are submitting.

The paper was carefully revised and a few changes have taken place. We have sent the revised manuscript. At the following, the points mentioned by the reviewer are discussed.

Reviewer:

The article is better than the first version.

Authors:

We appreciate the positive feedback from the reviewer. At this stage, we revised the content of subsection 2.1 in order to incorporate more challenging issues in quality control as reported in the studies [23-25]. This has brought considerable value to the literature review. Furthermore, it is also worth noting that the Conclusion section was carefully revised in order to explain the SWOT analysis (Figure 7) and various other improvements were made (e.g., English writing).

Thanks for the opportunity to revise the manuscript. We expect to fulfil your comments and suggestions.

With kind regards,

The manuscript authors,

Ramon Chaves
Daniel Schneider
António Correia
Claudia Motta
Marcos Borges